# Structural basis of human PCNA sliding on DNA

Matteo De March[1], Nekane Merino[2], Susana Barrera-Vilarmau[3], Ramon Crehuet[3], Silvia Onesti[1], Francisco J. Blanco[2,4] & Alfredo De Biasio[1]

Sliding clamps encircle DNA and tether polymerases and other factors to the genomic template. However, the molecular mechanism of clamp sliding on DNA is unknown. Using crystallography, NMR and molecular dynamics simulations, here we show that the human clamp PCNA recognizes DNA through a double patch of basic residues within the ring channel, arranged in a right-hand spiral that matches the pitch of B-DNA. We propose that PCNA slides by tracking the DNA backbone via a 'cogwheel' mechanism based on short-lived polar interactions, which keep the orientation of the clamp invariant relative to DNA. Mutation of residues at the PCNA–DNA interface has been shown to impair the initiation of DNA synthesis by polymerase δ (pol δ). Therefore, our findings suggest that a clamp correctly oriented on DNA is necessary for the assembly of a replication-competent PCNA-pol δ holoenzyme.

[1] Structural Biology Laboratory, Elettra-Sincrotrone Trieste S.C.p.A., 34149 Trieste, Italy. [2] CIC bioGUNE, Parque Tecnologico de Bizkaia Edificio 800, 48160 Derio, Spain. [3] Institute of Advanced Chemistry of Catalonia (IQAC), CSIC, Jordi Girona 18-26, 08034 Barcelona, Spain. [4] IKERBASQUE, Basque Foundation for Science, Bilbao 48013, Spain. Correspondence and requests for materials should be addressed to S.O. (email: silvia.onesti@elettra.eu) or to F.J.B. (email: fblanco@cicbiogune.es) or to A.D.B. (email: alfredo.debiasio@elettra.eu).

Processive chromosomal replication requires ring-shaped sliding clamp factors that encircle DNA and anchor polymerases and other proteins of the replisome. Proliferating cell nuclear antigen (PCNA)—the eukaryotic sliding clamp—is a homotrimeric ring of 86 kDa featuring a central channel lined with lysine and arginine-rich α-helices through which DNA is threaded[1–3]. The central channel of PCNA is a conserved structural feature among sliding clamps and is ~35 Å in diameter, significantly larger than the diameter of the B-form DNA double helix (~24 Å)[4]. Since the first determination of the structure of a sliding clamp, the bacterial β-clamp[5], this feature has raised questions regarding the interaction with DNA and the sliding mechanism. The crystal structure of β-clamp bound to primed DNA[6] showed the DNA duplex threaded through the clamp at a 22° tilt angle, making contacts with residues in the channel and on protruding loops on the back face of the clamp. However, extensive interactions are also established between the single-strand portion of DNA and the protein binding pocket of a crystallographically related β-clamp molecule. The β-clamp-ssDNA inter-molecular interaction in the crystal is intra-molecular in solution, and DNA competes with DNA polymerase binding[6].

Despite structural conservation, the absence of patterns of sequence similarity in bacterial and eukaryotic clamps[1] does not allow a simple correlation among the DNA-binding sites in the two systems. On the other hand, the positively charged

residues lining the clamp channel are conserved in eukaryotes and a subset of these residues function in DNA synthesis by pol δ and clamp loading by replication factor C (RFC)[7–9], pointing to the existence of direct interactions between PCNA and the DNA phosphodiester backbone. In a crystallographically derived model of yeast PCNA bound to a 10 bp primed DNA, the DNA in the clamp channel protrudes from the back face at a ~40° tilt angle, is held in place by a crystal contact, and would collide with PCNA if it was lengthened towards the front face. The PCNA–DNA interface only shows some charge complementarity and the protein residues involved largely do not overlap with those found to have an effect on clamp loading or pol δ function[7–9], making it difficult to ascribe this model to a functional state of the complex. Because of the inherent lability of the PCNA–DNA interaction, structural and biophysical characterization of this interface has been challenging. As a consequence, how eukaryotic clamps recognize DNA has remained controversial and a molecular mechanism of PCNA sliding on DNA has not been proposed. Here we have determined the crystal structure of human PCNA bound to a 10 bp long double-stranded DNA (dsDNA). We analysed the DNA-induced perturbations in the solution NMR spectrum of PCNA, and computationally characterized the interactions in multinanosecond molecular dynamics (MD) simulations. We show that PCNA recognizes the DNA structure through a set of basic residues within the ring channel organized to match the pitch of B-DNA, establishing

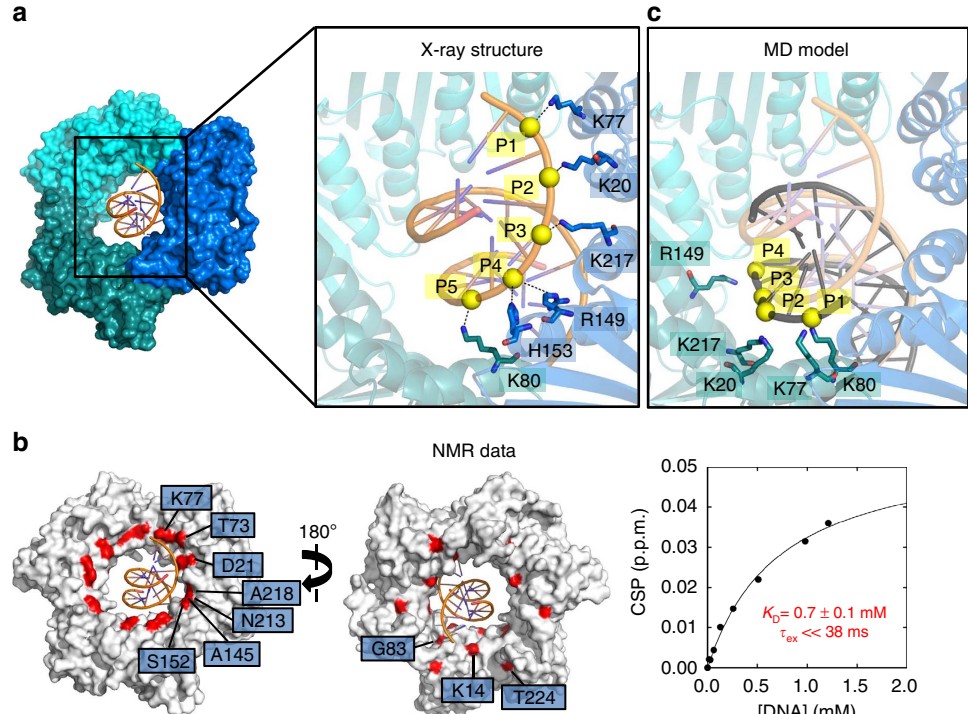

**Figure 1 | Structural basis of DNA recognition by human PCNA.** (**a**) 2.8 Å crystal structure of PCNA bound to a 10 bp DNA duplex. PCNA and DNA are shown in surface and ribbon representation, respectively. PCNA subunits are coloured in different shades of blue and DNA in orange. The expansion shows the complex in ribbon representation. Interacting PCNA side chains and DNA phosphates (interatomic side chain nitrogen—DNA phosphorus distance <5 Å) are shown as sticks and yellow spheres, respectively, and interactions as dashed lines (**b**) NMR analysis. Left: front- and back-face views of PCNA surface. PCNA residues whose amide chemical shifts are significantly perturbed by DNA are coloured red. The crystallographic position of DNA is shown in orange. The interacting region in the clamp channel overlaps with that seen in the crystal structure, whereas in the crystal, the side chains can be discriminated, in solution the perturbations involve the backbone amides. Right: chemical shift perturbation of the amide signal of PCNA residue T73 at different DNA concentrations. Fitting was performed using a single-site binding model. Extrapolated dissociation and exchange time constants are indicated. (**c**) Model interface from MD simulation. The crystallographic position of the DNA segment is shown in orange, whereas in black the DNA is shown in a position corresponding to the final state of the 100 ns MD simulation of the complex.

short-lived polar interactions with consecutive DNA phosphates. The interacting side chains are able to switch between adjacent phosphates in a non-coordinated manner, supporting a helical sliding mechanism in which the clamp rotates and tilts by keeping a fixed orientation relative to the DNA backbone. We discuss the implications of these findings for the function of the PCNA-pol δ holoenzyme in DNA replication, and also for clamp loading by RFC.

## Results

**Structure of the PCNA–DNA complex**. We obtained crystals of PCNA–dsDNA complex with one PCNA trimer per asymmetric unit and diffracting to 2.8 Å resolution (Supplementary Table 1). The dsDNA molecule threads through the PCNA ring with its longitudinal axis and the C3 axis of the ring forming an angle of 15° (Fig. 1a and Supplementary Fig. 1). No crystal lattice contacts between DNA and symmetry-related molecules are present, and the crystal packing does not affect the DNA position. The high temperature factors of dsDNA are compatible with a partial occupancy of dsDNA and/or with the existence of a subpopulation of complexes with slightly different DNA orientations. The complex interface involves the side chains of five basic residues (K20, K77, R149, H153 and K217), distributed on four α-helices of one PCNA subunit, and the side chain of

another residue (K80) on the proximal α-helix of the adjacent subunit. The side chains of the PCNA interfacial residues form a right-hand spiral that closely matches the pitch of B-DNA, and are involved in polar contacts with five consecutive phosphates of a single DNA strand (Fig. 1a).

The dynamic PCNA–DNA interface observed in the crystal is recapitulated by the NMR analysis of the binding in solution (Fig. 1b, Supplementary Figs 2a and 2b and 3). Titration of PCNA with dsDNA shows weak binding ($K_D \sim 0.7$ mM) to the inner side of the ring. In this assay, time averaging of the NMR signal restores the symmetry of the PCNA ring that is broken in the crystal. Backbone amide chemical shift perturbations are small (CSP < 0.06 p.p.m.), and the exchange rate is fast (Fig. 1b, Supplementary Figs 2 and 3). These observations are consistent with the low affinity of the interaction, the contacts involving long amino-acid side chains, and the high crystallographic temperature factors of DNA.

Using NMR, we tested the binding of PCNA to the primed-template DNA (pDNA) substrate (Supplementary Table 2) that was co-crystallized with the bacterial β-clamp[6]. In the β-pDNA structure, the ssDNA template strand is anchored to the main protein-binding pocket of the clamp, thus competing with DNA polymerase binding[6]. Our NMR mapping, however, shows that pDNA binds the inner wall of the human PCNA channel but not the protein binding (PIP-box) pocket

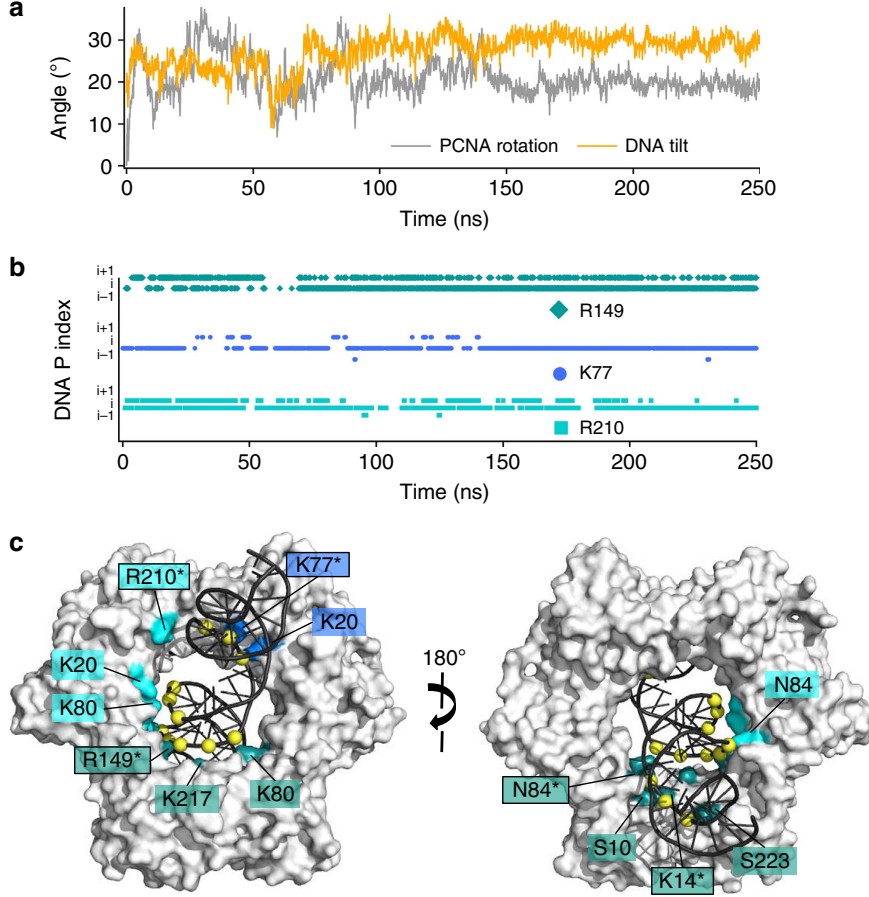

**Figure 2 | MD simulation of human PCNA bound to a 30 bp DNA.** (**a**) PCNA ring rotation and evolution of DNA tilting relative to the ring C3 axis (**b**) Time evolution of contacts between the side chain nitrogens of three representative PCNA interfacial residues and DNA phosphorus atoms at consecutive positions (**c**) Two views of the PCNA–DNA complex at the end of the MD trajectory. PCNA is shown as a grey surface and DNA as a black ribbon. PCNA residues whose side chains are engaged in polar contacts with DNA phosphates for >25% of the MD trajectory are labelled. Residues from different PCNA subunits are coloured in shades of blue. Residues that exchange between two or three consecutive DNA phosphates for >75% of the MD simulation are indicated by an asterisk and boxed.

(Supplementary Fig. 4), suggesting that bacterial and eukaryotic clamps recognize primed DNA differently.

**Dynamics of the PCNA–DNA interaction.** A 100 ns MD simulation of PCNA in complex with a 10 bp dsDNA predicts a highly dynamic interaction, in agreement with the experimental observations (Supplementary Figs 5a, 6 and 7). In the MD trajectory, DNA migrates from the crystallographic position to an alternative position, and subsequently to a central location with minimal interactions with the protein, to eventually collapse into a stable state (Supplementary Movie 1). Critically, the latter state can be obtained by shifting the crystallographic DNA model by two bases along the helical axis. Following this transformation, one DNA strand interacts with equivalent residues on the adjacent PCNA subunit (Fig. 1c). Distance analysis of the intermolecular polar contacts along the MD trajectory points to residues near the back face of the ring that contribute to driving DNA into its final state (S10, K14, N84; Supplementary Fig. 5b,c). These interactions are consistent with the NMR perturbation analysis (Fig. 1b). Thus, in solution DNA can access regions of the clamp that differ from, but are correlated to, those observed in the X-ray structure.

Altogether, these results anticipate that a DNA duplex longer than 10 bp will simultaneously bind two sets of B-helix matching residues on two PCNA subunits in a dynamic way. This prediction is supported by our 250 ns MD simulation of a complex of PCNA bound to a 30 bp dsDNA (Supplementary Table 2, Fig. 2, Supplementary Figs 6–9 and Supplementary Movie 2) which, having the fraying DNA duplex ends far from the ring, better recapitulates the physiological complex. The DNA at the end of the trajectory (Fig. 2a) shows extensive interactions with the clamp (Fig. 2c) and a more pronounced

tilting ($\sim$30°) compared with the crystallographic one. This value is slightly larger than the 20° reported for a 25 ns MD simulation[3]. In our simulation, however, 25 ns were found to be insufficient for the system to reach a stable conformation (Fig. 2a).

Single molecule diffusion data suggest that PCNA moves along DNA using two distinct modes: by rotationally tracking the DNA helix or, less frequently, by a faster motion uncoupled from the helical pitch[10]. Our MD simulations show that many of the PCNA interfacial residues can randomly switch between adjacent DNA phosphates on a sub-nanosecond time scale (Fig. 2b, Supplementary Fig. 9 and Supplementary Movie 3). This stochastic process will eventually generate a state where a sufficient number of contacts with adjacent phosphates in one direction of the helical axis are simultaneously established, resulting in a net rotation of the protein and the advancement of 1 bp (Fig. 3a). This 'cogwheel' mechanism would allow DNA backbone tracking in both directions while retaining DNA–protein contacts that keep the clamp in a defined orientation relative to DNA (Fig. 3b). On the other hand, the occasional exchange of DNA among the three equivalent positions of the PCNA homotrimer may account for the 'uncoupled' component of PCNA sliding.

## Discussion

The work presented here allows to visualize for the first time the atomic interactions between human PCNA and dsDNA, and to follow their time evolution. Our data support a molecular sliding mechanism that keeps the orientation of the clamp invariant relative to the DNA backbone. Importantly, we present results on the interaction of PCNA with primed DNA that reveal substantial differences compared to the bacterial system.

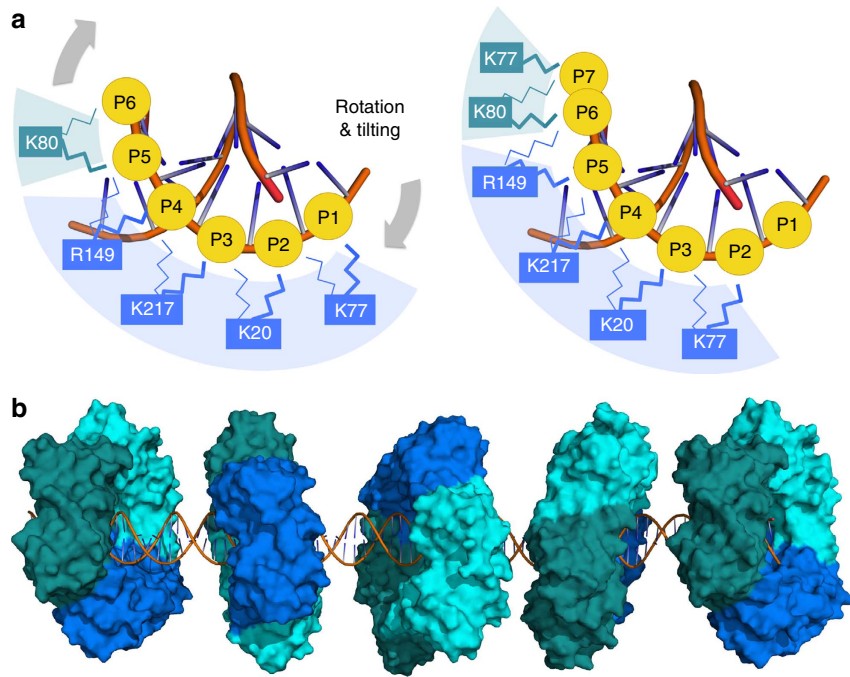

**Figure 3 | Proposed 'cogwheel' mechanism for PCNA sliding.** (**a**) Interacting side chains are able to rapidly switch between adjacent phosphates in a non-coordinated manner (illustrated by the thin and thick lines). When this stochastic process generates a state in which a sufficient number of electrostatic contacts are simultaneously established with adjacent phosphates in one direction of the DNA helical axis, a net rotation of the protein occurs and results in the advancement of one base pair (**b**) PCNA bi-directional tracking of the DNA backbone, which agrees with the clamp helical sliding mode inferred from diffusion data measured by single-molecule imaging[5]. The diffusion coefficient of PCNA (1.16 µm$^2$ s$^{-1}$) implies that, on average, PCNA diffuses 8 bp per microsecond.

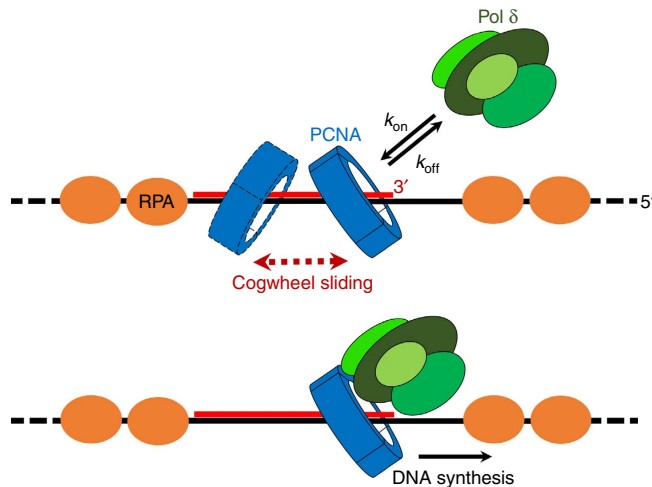

**Figure 4 | Possible role of PCNA–DNA interaction and sliding mechanism in lagging strand replication by pol δ.** PCNA is loaded on the P/T junction of a nascent Okazaki fragment by the clamp loader RFC. The RNA–DNA hybrid primer of the fragment (red) is blocked on both sides by Replication Protein A (RPA), which binds the template strand (black). The interaction of PCNA with DNA imparts an orientation to PCNA that is competent for pol δ binding and initiation of DNA synthesis. However, PCNA can slide off the P/T junction before binding pol δ or upon premature dissociation of pol δ from the template. The cogwheel sliding mechanism ensures that pol δ captures PCNA at the P/T junction of the fragment with the correct orientation to resume synthesis.

O'Donnell and co-workers[6] showed that the single-strand portion of primed DNA binds the protein binding pocket of β-clamp, and competes with DNA polymerase III binding. The data presented here, conversely, do not suggest an interaction between primed DNA and the polymerase-binding site of PCNA. Consequently, the functional models proposed for β-clamp based on this feature cannot be extended to eukaryotic clamps. In particular, the β-ssDNA contact may function as a 'placeholder' that keeps β-clamp near the 3′ end of primed DNA before polymerase binding and initiation of DNA replication[6]. Sticking of β-clamp to the 3′ end of primer/template (P/T) junctions of DNA was also observed by single-molecule experiments[11]. As we explain below, fast diffusion of human PCNA on DNA and our proposed sliding mechanism would bypass the need for such a placeholder function in DNA replication by the PCNA–pol δ complex.

Pol δ is a four-subunit B-family polymerase that replicates the DNA lagging strand by associating to PCNA, forming a holoenzyme at the P/T junction of nascent Okazaki fragments[12,13]. Three of the four human pol δ subunits interact directly with PCNA, via PIP-boxes that may bind up to three sites on the front face of the PCNA homotrimer[14–19]. Earlier evidence showed that a single mutation among PCNA residues K20, K77, K80, R149 and K217 severely reduces pol δ ability to incorporate an incoming nucleotide at the initiation of DNA synthesis[7]. Data presented in this paper suggest that these residues are critical for PCNA–DNA recognition and for orienting the clamp on DNA. Thus, we propose that this orientation is needed for the assembly of a functional pol δ holoenzyme, able to initiate replication of a primed Okazaki fragment (Fig. 4). A recent report showed that, unlike the yeast homologue, human pol δ maintains a loose association with PCNA while replicating DNA, and suggested that a significant fraction of pol δ holoenzymes may dissociate before finishing an Okazaki fragment[20]. If pol δ dissociates prematurely from the lagging strand template,

PCNA would be left behind on DNA for some time, where it would slide freely, until an incoming pol δ molecule rebinds to resume synthesis at the 3′ end of the aborted fragment. Thus, our proposed mechanism for PCNA sliding would ensure that the incoming polymerase encounters PCNA in the correct orientation to efficiently restart synthesis (Fig. 4). According to its diffusion coefficient ($\sim 1\,\mu m^2\,s^{-1}$)[10], free PCNA can slide over a fully formed Okazaki fragment (100–150 nt) in $< 0.2\,ms$, while nucleotide incorporation by pol δ is much slower ($k_{pol} \sim 100\,s^{-1}$)[20], suggesting that the sporadic sliding of PCNA off the P/T junction of an unfinished fragment would not significantly decrease the speed of replication. Likewise, pol δ dissociates from PCNA upon encounter of a DNA lesion, and PCNA is left behind before binding to a specialized translesion synthesis (TLS) polymerase able to replicate past the lesion[20]. Therefore, the proposed sliding mechanism of PCNA may as well be important for the assembly of functional complexes that involve TLS polymerases. Interestingly, the binding site of DNA on the inner wall of the PCNA ring partly overlaps with that of p15[PAF], an intrinsically disordered protein that regulates TLS via its interaction with PCNA[21–23]. Binding of p15[PAF] could modulate the PCNA sliding mechanism and dynamics, which may play a role in DNA repair.

A high-resolution crystal structure of a PCNA–polymerase–DNA ternary complex has not yet been determined, likely because of the inherent flexibility of the system. The medium-resolution electron microscopy structures of *Pyrococcus furiosus* (Pfu) PCNA bound to DNA and PolB or DNA ligase, determined by Morikawa and co-workers[24,25], show that the DNA duplex passing through PCNA is tilted. A recent computational work by Ivanov and co-workers[26] on the PfuPCNA–PolB–DNA complex show that the repositioning of the PolB core during the conformational switch from polymerizing to editing modes forces the DNA to tilt from one side of the PCNA channel to the other, suggesting flexibility of the PCNA–DNA interaction.

PCNA is loaded onto primer-template pDNA by the clamp loader RFC, a five-subunit complex that performs mechanical work through ATP binding and hydrolysis[27–30]. ATP binding enables the clamp loader to bind and open the clamp and bind pDNA, and ATP hydrolysis leads to the release of the clamp–pDNA complex, which can then associate to polymerases and other factors. Mutation of residues R14, K20, R80 or K217 in yeast PCNA (K14, K20, K80 and K217 in the human sequence) have been shown to slow binding of pDNA to the RFC–ATP–PCNA complex, slow clamp closure around pDNA after ATP hydrolysis, and hasten clamp slipping off pDNA[9]. Thus, these residues, which we show take part in the binding of human PCNA to DNA, may play a role in clamp loading by guiding DNA through the open clamp and into the clamp loader to form a tight complex. These residues may also assist in the transition of the clamp from open spiral to closed planar forms, hence promoting clamp–pDNA release from the clamp loader[27].

## Methods

**Protein expression and DNA duplexes.** Human PCNA (UniProt: P12004) was produced in *E. coli* BL21(DE3) cells grown in appropriate culture media to obtain protein with natural isotopic abundance or uniform enrichment using a clone with N-terminal His6-tag and PreScission protease cleavage site in a pET-derived plasmid. For NMR samples the protein was purified from the soluble fraction by $Co^{2+}$-affinity chromatography, cleaved by PreScission protease and polished by gel-filtration chromatography[31] in PBS (137 mM NaCl, 2.7 mM KCl, 10 mM sodium phosphate, 2 mM potassium phosphate) pH 7.0. All columns and chromatography systems used where from GE Healthcare. Protein elution was monitored by absorbance at 280 nm and confirmed by SDS–polyacrylamide gel electrophoresis. The purified protein contained the extra sequence GPH- at the N terminus. The PCNA sample for crystallization was obtained by introducing two additional purification steps[23]. The sample cleaved with PreScission protease was dialysed against 50 mM sodium acetate pH 5.5, 100 mM NaCl. After separation of

some precipitated material, the solution was loaded on a HiTrap Heparin HP column equilibrated with the same buffer. After column washing, the protein was eluted with a 0–100% gradient of 50 mM sodium acetate pH 5.5, 2 M NaCl in 20 column volumes (CV). The protein containing fractions of the major peak were dialysed against 20 mM Tris-HCl buffer pH 7.6, 150 mM NaCl and injected into a HiTrap Chelating HP column loaded with $Co^{2+}$ cations to remove uncleaved PCNA. The flowthrough was loaded on a HiTrap Q Sepharose column and eluted with a 0–60% gradient of 20 mM Tris-HCl pH 7.6, 1 M NaCl in 5 CV. The protein containing fractions were concentrated and polished using a Superdex 200 26/60 column equilibrated with PBS, pH 7.0, and then exchanged into the crystallization buffer (20 mM Tris-HCl, pH 7.5, 10% glycerol, 2 mM DTT) using a PD10 column. Stock solutions in PBS or crystallization buffer were flash-frozen in liquid nitrogen and stored at − 80 °C. The protein concentrations were measured by absorbance at 280 nm using the extinction coefficient calculated from the amino acid composition $(15,930 M^{-1} cm^{-1})$. All indicated concentrations of PCNA samples refer to protomer concentrations. dsDNA and pDNA duplexes were obtained by mixing equimolar amounts of the appropriate oligonucleotides in 20 mM Tris-HCl buffer pH 7.8, 25 mM NaCl, at 93 °C for 2 min with subsequent annealing by slow cooling at room temperature.

**PCNA–dsDNA complex structure determination.** Stocks of PCNA and dsDNA solutions were mixed to final concentrations of 0.4 mM and 1.1 mM, respectively (1:8 trimer:duplex molar ratio), and incubated at room temperature for 30 min before screening crystallization conditions by the hanging drop vapour diffusion method. Best diffracting co-crystals grew within 2 days at 20 °C in 2 µl droplets obtained by mixing 1 µl of the complex solution and 1 µl of a solution containing 11% polyethylene glycol 3350 in 0.1 M sodium acetate buffer, pH 4.5. Remote data collection was performed at the ID30A-3 (MASSIF-3) micro focus beamline (ESRF). Data processing and reduction was carried out with XDS[32,33] and the CCP4i suite[34]. Crystals belonging to space group H3 diffracted to 2.8 Å resolution and contained one PCNA trimer per asymmetric unit. The structure was determined by molecular replacement with MOLREP[33] using the previously published structure of the PCNA–p15[50–77] complex (PDB ID: 4D2G) as search model after removing the p15 peptide and solvent molecules. Repeated cycles of refinement using REFMAC5 (ref. 35) and model building with COOT[36] were performed. A global Non Crystallographic Symmetry restraint with TLS and Jellybody refinements were applied. For optimal modelling of bulk solvent within the ∼34 Å PCNA ring channel, Babinet's correction was applied[37]. The DNA within the PCNA channel was located in the centre of the ring at a 15° tilt angle by inspecting the unbiased Fo-Fc difference map contoured at 2.0–1.5σ (Supplementary Fig. 1). The DNA molecule was modelled and refined as a rigid body using a single TLS group. PDB_REDO[38] was used to check the quality of the crystallographic structure. Data collection and refinement statistics are listed in Supplementary Table 1. All figures with molecular models were prepared using PyMOL (www.pymol.org).

**NMR spectroscopy.** $^1H$–$^{15}N$ TROSY spectra were recorded at 35 °C on a Bruker Avance III 800 MHz (18.8T) spectrometer equipped with a cryogenically cooled triple resonance z-gradient probe. A 400 µl sample of 100 µM U-[$^2H,^{13}C,^{15}N,$] PCNA in 20 mM sodium phosphate buffer, 50 mM NaCl, pH 7.0, 20 µM 2,2-dimethyl-2-silapentane-5-sulfonate, 0.01% $NaN_3$ and 5% $^2H_2O$ was placed in a 5 mm Shigemi NMR tube (without plunger) and increasing volumes of DNA stock solutions were added and mixed (by capping and inverting the tube). The DNA stocks solutions were prepared as the PCNA samples in the same buffer (except that no $NaN_3$, DSS or $^2H_2O$ was added). For that purpose, and to remove unwanted salts from the synthetic oligonucleotides, they were dissolved in 20 mM sodium phosphate buffer, 50 mM NaCl, pH 7.0 (at a concentration between 0.7 and 4.9 mM) and desalted on a PD-10 Minitrap G25 column. For duplex formation, equimolar amounts were mixed and annealed (2 min at 95 °C in a thermoblock followed by slow cooling down to room temperature). The duplexes were concentrated by ultrafiltration up to 12.15 mM in 44 µl (dsDNA) or 11.21 mM in 71 µl (pDNA). These volumes were added stepwise to the PCNA samples, causing a 10 and 15% PCNA dilution, respectively. TROSY spectra were measured with 144 or 256 indirect points (alternating between 8 and 14 h total duration). The PCNA–dsDNA sample remained clear during the 5-day long titration at 35 °C, but there was an overall decreased in TROSY signal intensity (by about 25% measured in the most intense signal, the sharp signal of the C-terminal residue) and increased relative intensity at the central region of the spectrum (coming from both weak new sharp signals and background broad signals). This suggests that the PCNA protein was slowly losing structural integrity. The DNA duplex, however, remained homogeneous as assessed from the imino signals observed in one-dimensional proton spectra. In the case of pDNA titration, minor protein precipitation started after the 1:16 ratio addition, but the overall intensity decrease at the end of the titration (as measured on the C-terminal residue signal) was of the same order as in the case of dsDNA. This suggests that the largest contribution to the intensity loss along the titration is the binding of the large and protonated DNA duplexes to the deuterated PCNA ring. The pH of the PCNA samples was measured at the beginning and at the end of the titrations inside the NMR tubes and found to deviate by <0.1 units. Therefore, the small measured shifts are not caused by differences in pH or ionic strength. The small and steady shifts allowed for an extensive transfer of NMR signal assignments from the free PCNA to the DNA-bound PCNA spectra (with a coverage of 90% of non proline residues). The CSP values were computed as the weighted average distance between the backbone amide $^1H$ and $^{15}N$ chemical shifts in the free and bound states[39].

**MD simulations.** Two different systems were set up. First, a PCNA–DNA complex with the 10 bp DNA duplex, analogous to the system used in the crystallographic and NMR studies. The initial coordinates for this system were taken from the crystallographic structure (PDB ID: 5L7C). Second, a PCNA–DNA complex with the 30 bp DNA duplex. The coordinates of this model were based on the previous one for PCNA and the central 10 bp DNA segment. Then, the DNA chains were extended by 10 bp with B-form geometry along each direction using COOT[36]. The following steps are the same for both systems. First, the system was protonated with standard protonation states with Ambertools 15 (ref. 40), and solvated in a truncated dodecahedron box at least 1.5 nm away from the DNA or protein atoms. Chlorine and sodium ions were added to the simulation box to achieve a concentration of 100 mM and neutralize the system. Then, the system was minimized and equilibrated for 100 ps in the NVT ensemble and 100 ps in the NPT ensemble. Energy equilibration was checked for these steps. We run these initial steps with position constraints on DNA and protein heavy atoms. Then, we run a production simulation of 100 ns for the 10 bp system and 250 ns for the 30 bp system. When calculating averages, the first 10 ns (50 ns) were treated as equilibration and not considered for the 100 ns (250 ns) simulation. The stability of the simulations was checked by visual inspection of the trajectories and the RMSD with respect to the starting structure as plotted in Supplementary Fig. 6. Supplementary Figs 7 and 8 show that the DNA fragment remains in its double strand form and has a small curvature. We used the recently developed parmBSC1 force field[41], and TIP3P for the water model. All calculations were run with Gromacs 5 (refs 42,43). The superposition of structures and the calculation of the root mean square fluctuations (RMSF) were done with the Theseus Maximum Likelihood algorithm[44]. Theseus down-weights variable regions of the superposition and corrects for correlations among atoms, producing much more accurate results, especially for proteins having rigid and flexible regions. The analysis of DNA H-bonds and curvature (Supplementary Figs 7 and 8) was performed with 3DNA[45] and do_x3dna[46].

**Data availability.** Coordinates and structure factors of the PCNA–dsDNA complex are deposited in the Protein Data Bank under accession code 5L7C. The data that support the findings of this study are available from the corresponding author upon request.

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

## Acknowledgements

We thank M. Polentarutti and N. Demitri (Elettra Sincrotrone Trieste) for their assistance at the XRD1 beamline, N.G. Abrescia (CIC bioGUNE) and R. Steiner (King's College, London) for their advice in the crystallographic analysis, and A. Costa (Francis Crick Institute) for his useful comments. This work was supported by the Italian Association for Cancer Research (AIRC iCARE Fellowship to A.De.B. and AIRC Grant IG14718 to S.O.), and the Spanish Ministry of Economy and Competitiveness (CTQ2014-56966-R grant to F.J.B.). S.B.-V. acknowledges a fellowship from the Spanish Ministry of Economy and Competitiveness (Ref. BES-2013-063991). We acknowledge CERIC-ERIC for the use of the XRD1 beamline at Elettra Sincrotrone Trieste. The authors thankfully acknowledge the computer resources, technical expertise and assistance provided by the Red Española de Supercomputación, the Barcelona Supercomputing Center and the Catalan CSUC.

## Author contributions

A.D.B. designed the study. A.D.B., S.O. and F.J.B. guided the research experiments. N.M. purified the proteins. M.D.M. crystallized the complex, solved and refined the crystal structure. S.B.-V. and R.C. performed and analysed the MD simulations. F.J.B. performed and A.D.B. analysed the NMR experiments. M.D.M. prepared the manuscript figures. A.D.B. wrote the manuscript with contributions from all other authors.

## Additional information

**Competing financial interests**: The authors declare no competing financial interests.

