## [Peer Review File · Nature Communications]

Reviewers' Comments:

Reviewer #1 (Remarks to the Author)

In this study, De March et al analyzed the atomic interactions between PCNA and DNA by a combination of x-ray crystallography, NMR experiments, and MD simulations. The main conclusion is the presence of a dynamic interface, where the backbone atoms of DNA forms short-lived interactions with a set of basic residues facing the internal wall of the protein ring. These results are consistent with a sliding mode of PCNA that tracks the helical structure of DNA by binding to successive phosphate groups. These results are novel and well-supported by the data presented in the manuscript. I have only some minor comments:

1) In the Method section, it is stated: "The stability of the simulations was checked by visual inspection of the trajectories and the RMSD with respect to the starting structure as plotted in Fig S5". However the RMSD is not shown in figure S5 (nor in Figure 2 for the 30 bp system). These data should be included in the manuscript. Moreover, as the convergence of MD trajectories is always an issues for complex biological molecules, I suggest the authors to include any other data that could prove the convergence of the atomic trajectories. For instance, it could be interesting to show how the characteristics of the protein-DNA complex evolve along the MD trajectories (e.g. number of protein-DNA contacts, number of water molecules at the protein-DNA interfaces, geometrical features of the DNA double helix).

2) "The complex interface involves the side chains of five basic residues (K20, K77, R149, H153 and K217) distributed on four α -helices of one PCNA subunit and the side chain of another residue (K80) on the proximal α -helix of the adjacent subunit." A figure showing these interactions would help.

3) Table S2 is mentioned before Table S1

4) There is a typo on page 4 ("Table 1S")

5) Labels on Figure S6 are impossible to read. The authors should find a better way to represent those data, or remove the figure

Reviewer #2 (Remarks to the Author)

De March et al investigate how the sliding clamp PCNA interacts with DNA using NMR, X-ray crystallography and MD simulations. As has been seen before in two crystal structures (Georgescu et al Cell 2009 and McNally et al BMC Str. Bio. 2010), they observe that the DNA is tipped from being perpendicular to the plane of the ring. These results also match predictions based on MD simulations of PCNA and DNA (Ivanov et al NAR 2008). They propose a model for how the sliding clamp diffuses along DNA by tracking the helical pitch of DNA, which was previously shown by single molecule studies (Kochaniak et al JBC 2009 & Laurence et al JBC 2008 (not referenced in the manuscript)).

The data described in De March et al are strong and thorough.

However, the analysis appears to be a reprise of previous studies rather than groundbreaking research. The addition of NMR is nice but still is an incremental advance. I therefore recommend that the authors publish this research in another venue more appropriate for these types of findings.

I have some more minor issues with presentation of data and analysis, as delineated below:

On page 2 the authors state: "The high temperature factors of dsDNA are compatible with a partial occupancy of dsDNA and/or with the existence of a subpopulation of complexes with slightly different DNA orientations." Did the authors try to refine DNA occupancy?

The following sentence states: "The complex interface involves the side chains of five basic residues (K20, K77, R149, H153 and K217) distributed on four α -helices of one PCNA subunit and the side chain of another residue (K80) on the proximal α -helix of the adjacent subunit. The spatial arrangement of the interfacial PCNA residues matches the pitch of the DNA B-helix". However, these residues only interact with one strand of DNA. How then does this show specific contacts for dsDNA?

The authors later state: "The relatively high temperature factors of the contacting side chains indicate that they can interact with the DNA backbone in multiple conformations." What are the normalized b-factors of these residues? How do they compare with other residues in the pore? Ref. 3 is not a good reference for showing that DNA is threaded through the pore of the PCNA ring. This seems like the authors are simply trying to self-reference their own papers.

Reviewer #3 (Remarks to the Author)

In this work, De March et al. studied how PCNA interacts with DNA and slides on it using X-ray crystallography, NMR spectroscopy, and molecular dynamics (MD) simulations. The authors

found that several basic side chains interact with DNA within the ring channel of the PCNA trimer. Interestingly, these interactions occur in an asymmetric manner. By NMR-based titration experiment, these protein-DNA interactions within the channel were confirmed and the apparent dissociation constant for the complex was determined. Using the crystal structure, the authors also conducted MD simulations for the same complex as well as for a complex with extended DNA. Based on these results, the authors propose a mechanism for sliding of PCNA on DNA. Although I find this work very interesting, I have some major concerns as outlined below.

1. The biological importance of the findings from this work is not clear. PCNA play important roles for DNA repair and replication. However, sliding itself is basically one-dimensional diffusion that any DNA-binding proteins can undergo. Dynamic interactions between basic side chains and DNA phosphate groups, which are described in this manuscript, can occur in any nonspecific protein-DNA complexes. How does the proposed sliding mechanism contribute to PCNA's function?

2. The structural information about the PCNA-bound DNA duplex does not seem to be very reliable. The crystallographic B-factors for the 10-bp DNA are very high (Table S2) and in the crystallographic refinement, the DNA was refined as a rigid body using a single TLS group.

3. Another research group previously used MD simulations to study the PCNA-DNA interactions (Ref. 8). It is not clear from the current manuscript what are fundamentally new findings from the longer MD simulations in the current work.

Reviewer #1

In this study, De March et al analyzed the atomic interactions between PCNA and DNA by a combination of x-ray crystallography, NMR experiments, and MD simulations. The main conclusion is the presence of a dynamic interface, where the backbone atoms of DNA forms short-lived interactions with a set of basic residues facing the internal wall of the protein ring. These results are consistent with a sliding mode of PCNA that tracks the helical structure of DNA by binding to successive phosphate groups. These results are novel and well-supported by the data presented in the manuscript. I have only some minor comments:

1) In the Method section, it is stated: "The stability of the simulations was checked by visual inspection of the trajectories and the RMSD with respect to the starting structure as plotted in Fig S5". However the RMSD is not shown in figure S5 (nor in Figure 2 for the 30 bp system). These data should be included in the manuscript. Moreover, as the convergence of MD trajectories is always an issues for complex biological molecules, I suggest the authors to include any other data that could prove the convergence of the atomic trajectories. For instance, it could be interesting to show how the characteristics of the protein-DNA complex evolve along the MD trajectories (e.g. number of protein-DNA contacts, number of water molecules at the protein-DNA interfaces, geometrical features of the DNA double helix).

Three new supplementary figures have been included in the revised version reporting different parameters along the MD trajectories. Supplementary Fig. 6 shows the Root Mean Square Deviation (RMSD) of the complex and its components, whereas the stability of the geometric features of the DNA double helix are illustrated in Supplementary Fig. 7, showing the number of hydrogen bonds between the DNA base pairs, and Supplementary Fig. 8, showing the curvature of the 30 bp DNA.

Because PCNA will eventually slide along DNA, the MD simulations cannot be globally equilibrated. But the results shown in Fig. 2, Supplementary Fig. 6, 7 and 8 confirm that there is not a drift towards any unexplored structure, and that the simulations sample local conformations around a stable structure of the complex.

Figure S6 in the original version of the manuscript (now Supplementary Fig. 9) has been enlarged by splitting it into different panels, so as to display more clearly how the contacts of the interfacial protein residues with individual DNA backbone phosphates are exchanged with neighbouring phosphates along the simulation. This figure illustrates the highly dynamic nature of these interactions within the stable global conformation.

2) "The complex interface involves the side chains of five basic residues (K20, K77, R149, H153 and K217) distributed on four α -helices of one PCNA subunit and the side chain of another residue (K80) on the proximal α -helix of the adjacent subunit." A figure showing these interactions would help.

The orientation of the structure in Figure 1a has been changed and the PCNA–DNA interactions highlighted. The orientation of the structures in Figures 1b, 1c, 2c and Supplementary Fig. 5 has been modified accordingly.

3) Table S2 is mentioned before Table S1
This mistake has been corrected.

4) There is a typo on page 4 ("Table 1S")
This typo has been corrected.

5) Labels on Figure S6 are impossible to read. The authors should find a better way to represent those data, or remove the figure
Figure S6 (now Supplementary Fig. 9) has been split into panels and can now be inspected more easily.

Reviewer #2:

De March et al investigate how the sliding clamp PCNA interacts with DNA using NMR, X-ray crystallography and MD simulations. As has been seen before in two crystal structures (Georgescu et al Cell 2009 and McNally et al BMC Str. Bio. 2010), they observe that the DNA is tipped from being perpendicular to the plane of the ring. These results also match predictions based on MD simulations of PCNA and DNA (Ivanov et al NAR 2008). They propose a model for how the sliding clamp diffuses along DNA by tracking the helical pitch of DNA, which was previously shown by single molecule studies (Kochaniak et al JBC 2009 & Laurence et al JBC 2008 (not referenced in the manuscript)).

The data described in De March et al are strong and thorough.

We thank the reviewer for acknowledging the strength and completeness of our data.

However, the analysis appears to be a reprise of previous studies rather than groundbreaking research. The addition of NMR is nice but still is an incremental advance. I therefore recommend that the authors publish this research in another venue more appropriate for these types of findings.

The interaction of sliding clamps with DNA has been the focus of several studies, but how eukaryotic sliding clamps recognize DNA remained controversial. Below, we briefly revise the relevant papers quoted by the reviewer, highlighting the novelty of our findings with respect to the insights provided by these works:

1) X-ray derived model of *S. cerevisiae* PCNA bound to a 10 bp primed DNA (McNally et al, BMCSB, 2010)

This model was obtained from crystals of a non-native single-chain construct of yeast PCNA, in which the three PCNA monomers were fused together by two 11-residue linkers, and that contained mutations R110S and Y114S in the first subunit. Only 4 base pairs of the DNA duplex were visible in the unbiased electron density map calculated before DNA was placed in the model. The modelled DNA in the clamp channel protrudes from the back face at a $\sim 40^\circ$ tilt angle, is held in place by a crystal contact, and would collide with PCNA if it was lengthened towards the front face. The PCNA-DNA interface only shows some charge complementarity, no matching with the B-helix, and the protein residues involved largely do not overlap with those found to have an effect on clamp loading or polymerase δ function (Fukuda et al., JBC, 1995; McNally et al., BMCSB, 2010, Zhou et al., JBC, 2012). Because of these limitations, this model cannot be ascribed to a functional state of the complex, and falls short of providing mechanistic insights into PCNA sliding on DNA.

2) Crystal structure of *E. coli* β -clamp bound to a 10 bp primed DNA (Georgescu et al., Cell, 2008)

This structure shows the double stranded portion of DNA within the clamp channel at a $\sim 22^\circ$ tilt angle, making contacts with residues in the channel and on protruding loops on the back face of the clamp. However, extensive interactions are also established between the single-strand portion of DNA and the protein

binding pocket of a crystallographically-related β -clamp molecule. As the authors say, this crystal contact “*may also contribute to the observed angle of DNA*”. The authors show biochemically that the β -clamp/ssDNA inter-molecular interaction in the crystal is intra-molecular in solution. However, the atomic details of this intra-molecular interaction (which we show is not observed in human PCNA) and the relative DNA orientation are not described. The proposed model of the β -clamp-DNA complex in solution is consistent with a clamp trapped onto primer-template DNA, which may correspond to a snapshot of the clamp loading reaction, rather than the state of free sliding of the clamp on duplex DNA. This prevents the use of the model to infer a mechanism of sliding of β -clamp on dsDNA. Moreover, the differences among prokaryotic and eukaryotic clamps are such that results from one system cannot easily be extrapolated to the other.

3) *MD simulation of human PCNA in complex with a 28 bp DNA (Ivanov et al, NAR, 2006)*

The starting model used in this MD simulation does not rely on previous structural information on the PCNA-DNA interaction, and the simulation starts with DNA at an arbitrary position in the centre of the clamp pore, with the helical axis perpendicular to the ring plane. The trajectory ends with DNA tilted by $\sim 20^\circ$ and making interactions with the basic residues lining the clamp channel, but the exact PCNA and DNA residues participating in the binding are not reported. Competition between charged residues on the inner surface of PCNA for forming interactions with the two DNA strands was observed, but the identity of the competing residues not stated. Because of the short simulation time (25 ns), the authors did not attempt to propose a molecular mechanism for clamp sliding.

4) *Single-molecule imaging of clamp sliding on DNA (Kochaniak et al., JBC, 2009, and Laurence et al., JBC, 2008)*

In the work by Kochaniak and colleagues, diffusion coefficients are extrapolated from the trajectories of individual fluorescently-labelled human PCNA molecules loaded on stretched DNA. The diffusion data are consistent with a rotational movement of the protein around DNA, coupled to a much less frequent translational movement. In the authors’ own words: “*Even though our experiments indicate the existence of helical and non-helical diffusion modes of PCNA along DNA, we can only speculate on the underlying molecular mechanisms*”. Indeed, a helical motion around DNA may occur through different helical paths traced by the protein.

We did not quote the work by Laurence and colleagues on single-molecule tracking of bacterial β -clamp on DNA because, in that work, helical and non-helical diffusion modes of β -clamp are not resolved. However, this reference is now included and briefly discussed in the context of β -clamp binding to primed DNA.

In our work: 1) We report the first crystal structure of native human PCNA bound to duplex DNA, 2) The clamp-DNA interaction is not affected by crystal lattice contacts and is validated by solution NMR studies, 3) The binding interface involves several PCNA residues previously found to be important for clamp loading and polymerase δ function, 4) The 250 ns MD trajectory with the 30 bp DNA starts with DNA in the crystallographic position and reaches a stable configuration after ~ 100 ns, when DNA is tilted by $\sim 30^\circ$ and PCNA interfacial

residues exchange between consecutive DNA phosphates. These findings allow us to propose a molecular mechanism for DNA backbone tracking.

For these reasons we believe that the present work constitutes a major advance in the mechanistic understanding of PCNA sliding on DNA and its functional implications in DNA replication. In our work, we argue that the interaction of PCNA with DNA and the proposed mechanism of sliding have relevant implications in at least two processes intimately connected to PCNA function, namely initiation of DNA replication by polymerase δ , and clamp loading on DNA by replication factor C. The functional implications of our proposed sliding mechanism are described in detail in the current version of the manuscript.

I have some more minor issues with presentation of data and analysis, as delineated below:

1) On page 2 the authors state: "The high temperature factors of dsDNA are compatible with a partial occupancy of dsDNA and/or with the existence of a subpopulation of complexes with slightly different DNA orientations." Did the authors try to refine DNA occupancy?

As temperature factors and occupancies are highly correlated at this resolution, we have attempted to set the DNA occupancy to various values between 0.4-1.0 and refine the temperature factors; however this did not result in lower B-values or better Rfactor/Rfree statistics. We have also modeled an ensemble of 5 alternative DNA conformations as rigid bodies, characterized by small differences in the rotation angle about the helical axis ($\sim 10^\circ$) and tilting angle relative to the C3 axis of the PCNA ring ($\sim 3^\circ$), and set an occupancy of 0.2 for each position. In this case, B-factors were marginally lower, but Rfactor/Rfree was worse (0.26/0.29) and the quality of the electron density map appeared poorer. We can provide these data if needed.

In light of this, we decided to model the single DNA conformation (as a rigid body at full occupancy, albeit with high B factors) that yielded the best refinement statistics and electron density map quality.

2) The following sentence states: "The complex interface involves the side chains of five basic residues (K20, K77, R149, H153 and K217) distributed on four α -helices of one PCNA subunit and the side chain of another residue (K80) on the proximal α -helix of the adjacent subunit. The spatial arrangement of the interfacial PCNA residues matches the pitch of the DNA B-helix". However, these residues only interact with one strand of DNA. How then does this show specific contacts for dsDNA?

The last part of the sentence has been modified as follows:

"The side chains of the PCNA interfacial residues are arranged in a right-hand spiral that closely matches the pitch of B-DNA, and form polar contacts with five consecutive phosphates of a single DNA strand"

Because of the involvement of only one DNA strand, we do not claim specific contacts between PCNA and dsDNA based on our crystal structure. However, the PCNA residues at the crystallographic interface are organized to interact with

consecutive phosphates on a nucleotide chain in the right-handed helical conformation found in double stranded B-DNA. Figure 1a of the manuscript has been modified in order to highlight this complementarity.

3) The authors later state: "The relatively high temperature factors of the contacting side chains indicate that they can interact with the DNA backbone in multiple conformations." What are the normalized b-factors of these residues? How do they compare with other residues in the pore?

The overall B-factor estimated from Wilson Plot is 50.7 \AA^2 . The average B-factor for all the side chain atoms is 63 \AA^2 and the residues in the pore do not significantly differ from the overall value (64 \AA^2). The B-factor for the side chain of the residues that contact DNA is indeed higher (75 \AA^2), but probably the difference is not significant enough to justify our original statement. We have therefore deleted the sentence from the paper. We thank the reviewer for prompting us to carry out a more quantitative analysis on this issue.

4) Ref. 3 is not a good reference for showing that DNA is threaded through the pore of the PCNA ring. This seems like the authors are simply trying to self-reference their own papers.

Ref. 3 has now been substituted with Ivanov et al., NAR, 2006.

Reviewer #3:

In this work, De March et al. studied how PCNA interacts with DNA and slides on it using X-ray crystallography, NMR spectroscopy, and molecular dynamics (MD) simulations. The authors found that several basic side chains interact with DNA within the ring channel of the PCNA trimer. Interestingly, these interactions occur in an asymmetric manner. By NMR-based titration experiment, these protein-DNA interactions within the channel were confirmed and the apparent dissociation constant for the complex was determined. Using the crystal structure, the authors also conducted MD simulations for the same complex as well as for a complex with extended DNA. Based on these results, the authors propose a mechanism for sliding of PCNA on DNA. Although I find this work very interesting, I have some major concerns as outlined below.

1) The biological importance of the findings from this work is not clear. PCNA play important roles for DNA repair and replication. However, sliding itself is basically one-dimensional diffusion that any DNA-binding proteins can undergo. Dynamic interactions between basic side chains and DNA phosphate groups, which are described in this manuscript, can occur in any nonspecific protein-DNA complexes. How does the proposed sliding mechanism contribute to PCNA's function?

In this work, we argue that the described interaction of PCNA with DNA and the proposed mechanism of sliding have relevant implications in at least two processes: (a) initiation of DNA replication by polymerase δ (Pol δ), and (b) clamp loading on DNA by replication factor C (RFC).

(a) Pol δ replicates the DNA lagging strand by associating to PCNA at the primer/template junction of nascent Okazaki fragments. Earlier evidence (Fukuda et al., JBC, 1995) showed that a single alanine mutation among the five PCNA basic residues present in the binding site of our crystal structure (K20, K77, K80, R149 and K217) dramatically reduced Pol δ capacity to incorporate an incoming nucleotide. Rather than affecting Pol δ processivity, these mutations seemed to impair the initiation of DNA synthesis by Pol δ . This strongly suggested that the specific interaction of PCNA with the threaded DNA duplex impacts the function of the Pol δ holoenzyme. Based on our results, we argue that PCNA does not interact with the DNA backbone randomly, but rather that a specific set of polar contacts is required to keep PCNA in a defined orientation relative to the phosphodiester backbone. We propose that this orientation is needed for the assembly of a functional PCNA-Pol δ -DNA complex, able to initiate replication of a primed Okazaki fragment.

A recent report shows that human Pol δ maintains a loose association with PCNA while replicating DNA, and suggests that a significant fraction of Pol δ holoenzymes may dissociate before finishing an Okazaki fragment (Hedglin et al., PNAS, 2016). If Pol δ dissociates prematurely from the lagging strand template, PCNA would be left behind on the DNA for some time, where it would slide freely, until an incoming Pol δ molecule rebinds to resume synthesis at the 3' end of the aborted fragment. We propose that the "cogwheel" mechanism for PCNA sliding, in which the protein rotates and tilts by keeping a fixed orientation

relative to the DNA backbone, ensures that the incoming polymerase finds PCNA in the correct position relative to DNA in order to resume synthesis. Likewise, Pol δ dissociates from PCNA upon encounter of a DNA lesion, and PCNA is left behind before it binds a specialized translesion synthesis (TLS) polymerase able to replicate past the lesion (Hedglin et al., PNAS, 2016). Therefore, we speculate that the sliding mechanism of PCNA may as well be important for the assembly of functional replication complexes that involve DNA repair polymerases.

(b) The PCNA clamp is loaded onto primer/template (pt) DNA by the clamp loader RFC, a five-subunit complex that performs mechanical work through ATP binding and hydrolysis. ATP binding enables the clamp loader to bind and open the clamp and bind ptDNA, and ATP hydrolysis leads to the release of the clamp-ptDNA complex, which can then associate to DNA polymerases and other factors. R14, K20, R80 and K217 mutations in *S. cerevisiae* PCNA (K14, K20, K80 and K217 in the human sequence) were shown to slow binding of ptDNA to the RFC-ATP-PCNA complex, slow clamp closure around ptDNA after ATP hydrolysis, and hasten clamp slipping off ptDNA (Zhou et al., JBC, 2012). Thus, these residues may play a role in clamp loading by guiding DNA through the open clamp and into the clamp loader to form a tight complex, and by assisting the transition of the clamp from open spiral to closed planar forms, thus promoting clamp-ptDNA release from the clamp loader. In our crystal structure of human PCNA-DNA complex, K20, K80 and K217 appear at the interface. Thus, we propose that these interfacial residues may be important in different steps of the clamp loading reaction to impart a specific direction to DNA.

These functional aspects of our findings are now described in detail in the discussion section of the current manuscript. A new figure (Fig. 4) illustrates the hypothesis on the relevance of PCNA orientation with respect the DNA on the initiation of DNA replication by polymerase δ .

2) The structural information about the PCNA-bound DNA duplex does not seem to be very reliable. The crystallographic B-factors for the 10-bp DNA are very high (Table S2) and in the crystallographic refinement, the DNA was refined as a rigid body using a single TLS group.

The reliability of our crystal structure is accurately expressed in a series of statistical parameters, that describe the quality of the diffraction data (Resolution, Rmerge, I/sI, completeness, redundancy) and the quality of the model (Rwork, Rfree and other refinement indices). As reported in Supplementary Table 1, all of these parameters indicate that the data collected from our crystals are complete, accurate and significant, and that the model fits well the data (Rwork and Rfree), has a good geometry (R.m.s. deviations) and acceptable temperature factors.

The temperature factors for the DNA are large but, as discussed in the paper, this is due to the intrinsic flexibility of the dsDNA within the channel, and the likely presence of multiple conformations, consistent with the biological function of the protein.

Indeed this is not unusual in the crystal structures of DNA binding proteins and enzymes that interact with DNA in a non-sequence specific fashion, and that establish transient interactions allowing rapid diffusion of the protein along the nucleic acid.

High B-factors were observed in the structure of yeast PCNA bound to primed DNA (PDB ID: 3K4X; McNally et al.) where the DNA was also refined as a rigid body with TLS and where the temperature factors range from 130 to 450 Å². The DNA is slightly more ordered in the structure of the bacterial β-clamp with primed DNA (PDB ID: 3BEP; Georgescu et al.), but here also the B-factors of numerous nucleotides reach 90-110 Å².

Numerous other examples include the structures of the T7 polymerase in complex with DNA (PDB ID: 1S76, 1S77; Yin et al., 2004, Cell), with B-factors of 120-130 Å²; the RuvA-Holliday junction complex (PDB ID 1C7Y; Ariyoshi et al., 2000, PNAS) with B-factors up to 150 Å², crystal structure of Mot1 and TBP bound to DNA, with B-factors ranging from 90 to 280 Å² (PDB ID: 4WZS; Butryn et al, 2015 eLife), the ssDNA bound to the channel of the E1 papillomavirus Helicase (PDB ID 2GXA; Enemark et al., 2006, Nature), with B-factors reaching 180 Å², the multidrug tolerance factor HipAB, with B-factors for the DNA in the range 140-220 Å² (PDB ID: 5K5Q; Schumaner et al., 2015, Nature), the histone-like protein HU (Hammel et al., to be published; PDB ID 4YEY), with DNA temperature factors from 150 to 220 Å²). From this long - but far from exhaustive - list, it is clear that although the B-factors of the dsDNA in our structure are high, they are not unusually high.

Additionally, to complement the crystallographic results we present further evidence from NMR and molecular dynamics studies. This is in line with the new paradigms at the forefront of structural biology, where an integrated approach is used to best describe the multifaceted behaviour of complex and dynamic biological systems that cannot easily be described by a single, static picture.

3) Another research group previously used MD simulations to study the PCNA-DNA interactions (Ref. 8). It is not clear from the current manuscript what are fundamentally new findings from the longer MD simulations in the current work.

In their MD simulation of PCNA bound to a 28 bp DNA, Ivanov and colleagues (Ivanov et al., NAR, 2006) used a model that could not rely on previous structural information on the PCNA-DNA interaction. Consequently, their MD starts with DNA in the centre of the clamp pore in an arbitrary position, with the helical axis perpendicular to the ring plane. The 25 ns trajectory ends with DNA tilted by ~20° and making interactions with the basic residues lining the clamp channel. However, the exact residues involved in the interaction along the trajectory are not reported. Competition between PCNA interfacial residues with the two DNA strands was observed, but the identity of the competing residues not stated. Because of the short simulation time, the authors did not attempt to propose a molecular mechanism for clamp sliding.

To perform our MD simulations of PCNA bound to (i) 10 and (ii) 30 bp DNA substrates, we built models based on our X-ray structure of the PCNA-DNA complex and we explored longer simulation times.

The 100 ns MD of complex (i) shows the highly dynamic nature of the interaction of PCNA with the short DNA duplex that was co-crystallized with PCNA. This MD trajectory allows visualizing the dynamic process of DNA binding to the second set of B-helix matching residues in the clamp inner wall, and helps interpret the NMR data showing interactions with PCNA residues located at the clamp back face.

The model for the 250 ns MD of complex (ii) features the central 10 bp DNA segment in the crystallographic position extended by 10 bp with B-form geometry along each direction. In this MD trajectory, the system reaches a stable configuration after ~ 100 ns, when DNA is tilted by $\sim 30^\circ$ and PCNA interfacial residues exchange between consecutive DNA phosphates.

The two simulations show essentially the same molecular recognition events between the DNA and PCNA, the difference in tilting angles being likely caused by the DNA length. The results from the simulations together with the NMR and X-ray experiments allowed us to propose the cogwheel mechanism for backbone tracking.

Reviewers' Comments:

Reviewer #1 (Remarks to the Author):

The authors satisfactorily answered all my comments. I think that this version of the manuscript is suitable for publication

Reviewer #2 (Remarks to the Author):

De March et al provide a much improved manuscript. They (mostly) address my previous concerns about novelty.

However, there remain some important concerns that should be addressed prior to publication.

1. The Authors state on line 48: 'No crystal contacts affect the clamp-DNA interaction.' This statement should be explained better. Specifically, are there no contacts at all between DNA and adjacent molecules in the crystal? What about crystal positioning that would sterically restrict the positioning of DNA in a perpendicular manner?

2. The authors state in lines 78-81 that the residues near the back face of the ring drive DNA positioning in MD and that these residues are consistent with NMR perturbation analysis. However, this is hard to see from the figures shown (NMR Supp FIg4 and MD in Fig 1b). Labels on Supp Fig4 would help asses consistency between the two results.

3. The last paragraph of results seems like Discussion to me.

4. The authors refer to the O'Donnell Beta clamp/DNA structure in lines 113-4: "Thus, the crystal structure may represent a snapshot corresponding to the termination of the clamp loading reaction, rather than the "sliding" state."

This statement doesn't make sense to me. If the loaded clamp is held by intramolecular interactions, then how is this state mimicked by the inter-molecular crystal packing forces? It seems that the orientation of DNA would likely be different if the ssDNA overhang made an intramolecular interaction than intermolecular. This statement should probably be removed.

5. In lines 139-140: "Here we show that these residues are essential for PCNA–DNA recognition and for orienting the clamp on DNA."

Actually this has not been shown. To demonstrate essentiality, the authors would need to mutate these residues and show that DNA orientation is disrupted. This statement needs to be amended

to reflect that this is a hypothesis.

6. The authors added a section discussing how the residues lining the pore could be important for maintaining position on DNA for binding of polymerases. I think that this is a nice addition. I'd like to point out that the Morikawa group has investigated the structure of PCNA bound enzymes by EM (Nishida et al & Mayanagi et al PNAS 2009). Their data suggest that both polymerases and ligases position DNA in a tilted orientation through PCNA. For DNA pol, it appears that the pol mode may position DNA perpendicular to the ring, but the exo mode uses the tilted orientation. Thus, PCNA may exhibit significant flexibility in terms of the orientation of DNA. I think that this data should be mentioned to as another prior observation of the titled binding mode.

7. The final paragraph is a bit odd. Are the authors insinuating that the PCNA/DNA interactions are what is driving the entire replisome to rotate? This seems like the tail wagging the dog. Of course DNA polymerases will rotate during DNA synthesis, even in the absence of PCNA; after all, the pol active site must track the helix of DNA by definition. Likewise for a helicase. I recommend removing this paragraph, as it is confusing and adds little to the manuscript.

Minor points:

1. The authors assert that the 'sidechains of the PCNA interfacial residues form a right-handed spiral that closely matches the pitch of B-DNA.'

Because the sidechains interacting with DNA are very flexible, could they adopt an A-form spiral? In other words, how specific is the PCNA for B-form DNA?

2. Label in Figure 1 has His152. Should be His153.

Reviewer #2 (Remarks to the Author):

De March et al provide a much improved manuscript. They (mostly) address my previous concerns about novelty.

However, there remain some important concerns that should be addressed prior to publication.

1. The Authors state on line 48: 'No crystal contacts affect the clamp-DNA interaction.' This statement should be explained better. Specifically, are there no contacts at all between DNA and adjacent molecules in the crystal? What about crystal positioning that would sterically restrict the positioning of DNA in a perpendicular manner?

Indeed, no contacts are present between DNA and adjacent molecules in the crystal. In addition, the crystal packing would not hinder a positioning of DNA perpendicularly to the PCNA ring plane. This can be easily seen by inspecting the PDB of our crystal structure.

For a clearer explanation, the statement in the text has been reformulated as follows: "No crystal lattice contacts between DNA and symmetry related molecules are present, and the crystal packing does not affect the DNA position"

2. The authors state in lines 78-81 that the residues near the back face of the ring drive DNA positioning in MD and that these residues are consistent with NMR perturbation analysis. However, this is hard to see from the figures shown (NMR Supp Fig4 and MD in Fig 1b). Labels on Supp Fig4 would help assess consistency between the two results.

We think the Reviewer refers to Supplementary Fig. 5, rather than Supplementary Fig. 4, as Fig. 1b (dsDNA binding shown by NMR) compares directly with Supplementary Fig. 5 (dsDNA binding shown by MD simulation) and not Supplementary Fig. 4 (pDNA binding shown by NMR). However, for clarity, labels on PCNA residues interacting with pDNA have been added to Supplementary Fig. 4.

3. The last paragraph of results seems like Discussion to me.

The last paragraph of Results describes the data from the MD simulation of PCNA binding to the 30 bp DNA, specifically the time evolution of contacts between PCNA side chains and DNA phosphate atoms. Although the sliding mechanism that we infer from these data is a proposition, it directly stems from the data. Therefore, we would prefer to keep this paragraph in the Results section.

4. The authors refer to the O'Donnell Beta clamp/DNA structure in lines 113-4: "Thus, the crystal structure may represent a snapshot corresponding to the termination of the clamp loading reaction, rather than the "sliding" state."

This statement doesn't make sense to me. If the loaded clamp is held by intramolecular

interactions, then how is this state mimicked by the inter-molecular crystal packing forces? It seems that the orientation of DNA would likely be different if the ssDNA overhang made an intramolecular interaction than intermolecular. This statement should probably be removed.

We agree with the Reviewer that the statement may not be correct, as the orientation of DNA in the crystal structure of β -clamp–DNA complex may be different from that in the complex in the solution state. The statement has therefore been removed from the text.

5. In lines 139-140: “Here we show that these residues are essential for PCNA–DNA recognition and for orienting the clamp on DNA.”

Actually this has not been shown. To demonstrate essentiality, the authors would need to mutate these residues and show that DNA orientation is disrupted. This statement needs to be amended to reflect that this is a hypothesis.

The statement has been modified as follows: “Our data suggest that these residues are critical for PCNA–DNA recognition and for orienting the clamp on DNA”

6. The authors added a section discussing how the residues lining the pore could be important for maintaining position on DNA for binding of polymerases. I think that this is a nice addition. I’d like to point out that the Morikawa group has investigated the structure of PCNA bound enzymes by EM (Nishida et al & Mayanagi et al PNAS 2009). Their data suggest that both polymerases and ligases position DNA in a tilted orientation though PCNA. For DNA pol, it appears that the pol mode may position DNA perpendicular to the ring, but the exo mode uses the tilted orientation. Thus, PCNA may exhibit significant flexibility in terms of the orientation of DNA. I think that this data should be mentioned to as another prior observation of the titled binding mode.

We thank the Reviewer for pointing out the work of Morikawa’s group. A paragraph has been added to the discussion mentioning these contributions.

7. The final paragraph is a bit odd. Are the authors insinuating that the PCNA/DNA interactions are what is driving the entire replisome to rotate? This seems like the tail wagging the dog. Of course DNA polymerases will rotate during DNA synthesis, even in the absence of PCNA; afterall, the pol active site must track the helix of DNA by definition. Likewise for a helicase. I recommend removing this paragraph, as it is confusing and adds little to the manuscript.

Our statement wanted to convey the idea that a rotational motion of PCNA around DNA matches the expected rotations of other components of the replisome during replication. We agree, however, that the statement may be confusing and we have therefore removed it from the text.

Minor points:

1. The authors assert that the ‘sidechains of the PCNA interfacial residues form a right-handed spiral that closely matches the pitch of B-DNA.’

Because the sidechains interacting with DNA are very flexible, could they adopt an A-form spiral? In other words, how specific is the PCNA for B-form DNA?

Because of the flexibility of the PCNA interacting side chains and the structural similarities between A- and B-form DNA, the interaction pattern seen in our crystal structure may as well apply to A-form DNA. However, the DNA in our models is in the B-form, and we therefore highlight the complementarity of the interacting residues with B-form DNA.

2. Label in Figure 1 has His152. Should be His153.

This mistake has been corrected.